# Optimisation of QCL Structures Modelling by Polynomial Approximation

**DOI:** 10.3390/ma15165715

**Published:** 2022-08-19

**Authors:** Stanisław Pawłowski, Mariusz Mączka

**Affiliations:** 1Department of Electrodynamics and Electrical Machine Systems, Faculty of Electrical and Computer Engineering, Rzeszow University of Technology, 35-959 Rzeszow, Poland; 2Department of Electronics Fundamentals, Faculty of Electrical and Computer Engineering, Rzeszow University of Technology, 35-959 Rzeszow, Poland

**Keywords:** Schrödinger–Poisson equations, semiconductor superlattices, polynomials approximation, transfer matrix method

## Abstract

Modelling of quantum cascade laser (QCL) structures, despite a regular progress in the field, still remains a complex task in both analytical and numerical aspects. Computer simulations of such nanodevices require large operating memories and effective algorithms to be applied. Promisingly, by applying semi-analytical polynomial approximation method to computing potential, wave functions and electron charge distribution, accurate results and quick convergence of the self-consistent solution for the Schrödinger and Poisson equations are reachable. Additionally, such an approach makes the respective numerical models competitively effective. For contemporary QCL structures, with quantum wells quite typically forming complex systems, a special approach to determining self energies and coefficients of approximating polynomials is required. Under this paper we have analysed whether the polynomial approximation method can be successfully applied to solving the Schrödinger equation in QCL. A new algorithm for determining self energies has been proposed and a new method has been optimised for the researched structures. The developed solutions have been implemented as a new module for the finite model of the superlattice (FMSL) and tested on the QCL emitting light in the mid-infrared range.

## 1. Introduction

QCL structures are nanodevices that have currently grown very popular among scientific teams around the world [1,2,3,4,5]. Their applications likely to be adapted in many important economic branches, such as spectroscopy [6], telecommunications [7], or medicine [8] to name a few, have driven continuous research, with computer simulations playing a major role [9,10,11,12,13,14]. The formalism of non-equilibrium Green’s functions has proven to be the most comprehensive approach for it. It can be applied in the form of a real space model (RSM) [15,16], or a model based on the properties of Wannier functions (WFM) [17,18]. The WFM model is faster, and like the RSM, it assumes infinite geometrical sizes of the researched structures. Due to the hardware limitations, simulations with the concerned models are restricted to either one (RSM), or three (WFM) periods of the structure, which, in fact, consists of twenty [19] to nearly two hundred [20] modules. In our previous papers [21,22,23,24] we showed the finite geometrical sizes of the superlattice model to have a significant impact on the simulations results. The finite superlattice model (FMSL) [24] we proposed, due to the semi-analytical approach to the self-consistent solution of the Schrödinger and Poisson equations, not only allowed us to incorporate any number of structural periods, but it was also found as effective as the WFM. The developed FMSL has proven very successful for simulating simple superlattice structures [25] and QCL-THz [20]. Nevertheless, while calculating QCL emitting mid- and far-infrared radiation [26,27], problems aroused. Such devices contain many narrow quantum wells, separated with relatively high potential barriers. As such, they generate quantum states vital for electronic transport in very narrow minibands, which are hard to detect. Therefore, it was necessary to significantly increase the model accuracy with respect to determining self energies by implementing additional program modules with new numerical algorithms incorporated. The approximation of wave functions resulting from the applied approach was another aspect, as in QCL structures under consideration it can take on very complicated forms. This requires checking and matching the degree of the approximating polynomial while simulating. As the newly developed software significantly extended the simulation time, efforts were focused on optimising the new version of the model.

Under this paper, numerical procedures expanding the FSML application to simulating the transport parameters for QCL emitting mid- and far-infrared radiation have been described and analysed. The solutions to the encountered problems are presented, as well as a significant progress in relation to the previously developed FMSL is shown, in particular, the increased accuracy and efficiency. In order to illustrate the effectiveness of the new model, the simulations results for representative QCL are presented.

As the paper deals with several quite separate issues, we divided it into following sections:-Chapter 2, where operation of QCL with a typical structure emitting radiation in the mid-infrared range is briefly described;-Chapter 3, where we present the numerical QCL model into which a new algorithm for precise self energy search has been incorporated, and its basic procedures are described;-Chapter 4, where validity of implementing such changes into the current QCL model is discussed and supported with relevant calculation results.

And last but not least

-Chapter 5, where the convergence of solutions to the Schrödinger equation is analysed, which is of key importance for model optimisation, as confirmed with relevant examples.

The performed studies have finally led to creating an efficient and accurate QCL model dedicated to systems with very narrow quantum wells and high potential barriers for electronic transport.

## 2. Quantum Cascade Laser

The structure of a quantum cascade laser can be represented as a periodic potential derived from nanometer layers of two different semiconductors arranged alternatively. The application of an electric field in such a system determines electron transport enriched with quantum phenomena, which can be used to emit radiation of different energy. The idea of such a mechanism is schematically illustrated in Figure 1 on the basis of the structure described in [28].

Each module of the laser structure consists of two parts, namely, the injector, and the active area. Three basic quantum states are available in the active area, namely, a high (c), medium (b), and low (a) state. The electron injected from the previous period of the structure into the state (c) is transferred into the state (b) by emitting a photon of the energy equal to the energy difference between subbands (c) and (b). Then, the carrier reaches the low state (a) and its energy is transferred to the crystal structure as a phonon. The injection area of the next period is designed to carry the electron from the low state (a) of the active region and enable its further transport to the high state (c’) of the next structural module. The presented sequence of transitions means that one electron is the source of many photons; hence, such lasers are characterised by high output powers.

For the correct operation of the device, it is necessary to ensure the population inversion for high quantum states, at a level that allows cascading photon emissions in subsequent periods of the structure. Unfavourable physical phenomena causing the escape of electrons into the energy continuum makes it much more complex. Therefore, the first cascade lasers worked properly only at cryogenic temperatures [29]. The solution to the problem was achieved by appropriately configuring two [30] or three [31] quantum wells, or even complete superlattices [32,33] within the active area of the laser structure. By introducing a very thin quantum well, followed by two wider quantum wells in the active area of the laser, we have managed to significantly increase the efficiency of injecting carriers to the upper level (c) and to reduce the effect of harmful transfer of carriers directly to the lower quantum level (a). All these solutions generate numerous quantum states, often very similar in energies and grouped in very narrow mini-bands. They can be determined by self-consistent solving the Schrödinger and Poisson equations, though it remains quite a difficult task. The problem can be solved in several ways, one of which is described in the following chapter.

## 3. The Model of the QCL

The simulations of the QCL structures were carried out in two stages. First, the Schrödinger and Poisson equations were solved self-consistently. Then, the base of quantum states obtained from such calculations was used in while finding the Green functions. On the basis of Green’s functions, it was possible to exactly determine the transport parameters characterising the quantum phenomena described in Section 2. The main simulation algorithm is illustrated in Figure 2, where we explicitly show an iterative loop of the self-consistent solution of the Schrödinger–Poisson equation controlled by the value of the $\delta_{FM\min}$ coefficient, as well as two program blocks responsible for generating the system quantum states base and further determination of Green’s functions necessary to calculate transport parameters for the structure. The convergence coefficient for the Schrödinger and Poisson equations was defined as
(1)$$\delta_{FM}=\sqrt{\frac{{\int \left(\rho_{k}-\rho_{k-1}\right)}^{2}dz}{\int \rho_{k}^{2}dz}}\cdot 100\%$$
where *ρ**_k_* represents the total charge density of the structure for the *k*th iteration. The algorithm carries out a loop in which both equations are solved until the charge changes reach the value of $\delta_{FM\min}$.

The numerical model used for the self-consistent solution of both equations is schematically illustrated in Figure 3. We can see there the potential distribution in the direction of electron transport for the QCL structure polarised with the voltage ***U***, along with a set of parameters describing it. In the area of each superlattice layer, a constant value of the effective mass of the electron and a variable potential described by the polynomial function *V_j_*(*z*) were assumed.

The QCL simulation process begins with solving the Schrödinger equation, as previously reported [24], reduced to a dimensionless form
(2)$$\frac{\partial^{2}\Psi }{\partial u_{}^{2}}+\left[1-W_{j}\left(u\right)\right]\;\Psi \left(u\right)=0$$

The following dimensionless variables were adopted here as
(3)$$W_{j}\left(u\right)=\frac{V_{j}\left(z\right)}{E}\;\mathrm{and}\;u_{j}=\frac{\sqrt{2m_{e}{}_{j}eE}}{\hbar }\left(z-z_{j}\right)$$
where *E*, *i*, and $m_{e}{}_{j}$ represent the energy and effective mass of the electron in the *j*th superlattice layer, respectively, while *e* stands for the elementary charge, and *V_j_* (*z*) for the total potential in the *j*th layer of the system defined as
(4)$$V_{j}\left(z\right)=V_{SLj}\left(z\right)+V_{Bj}\left(z\right)+V_{Sj}\left(z\right)$$
where *V_SLj_*, *V_Bj_*, and *V_S_*, are the potentials of the superlattice, applied voltage, and unbalanced dopant charge, respectively. Simultaneously, it was assumed that the total potential is represented by a polynomial, which leads to the relationship as below
(5)$$W_{j}\left(u\right)=\sum_{k=0}^{M}d_{j,k}u_{}^{k},\;d_{j,k}=\frac{b_{j,k}}{E\sigma_{j}^{k}},\;\sigma_{j}=\frac{\sqrt{2m_{e}{}_{j}eE}}{\hbar }$$

According to the proposed approach, the potential *V_j_* (*z*) is represented by a power series in the form
(6)$$V_{j}\left(z\right)=\sum_{k=0}^{M}b_{j,k}{\left(z-z_{j}\right)}^{k}$$
the boundary conditions for Equation (2) were assumed as
(7)$$\Psi_{j}\left(z_{j+1}\right)=\Psi_{j+1}\left(z_{j+1}\right)$$
(8)$$m_{e}{}_{j}{\frac{d\Psi_{j}}{dz}|}_{z_{j+1}}={m_{e}{}_{j+1}\frac{d\Psi_{j+1}}{dz}|}_{z_{j+1}}$$
while solutions were also sought in the form of a power series
(9)$$\Psi_{j}\left(u\right)=\sum_{n=0}^{\infty }c_{j,n}u_{}^{n}$$
where
(10)$$c_{j,n}=\frac{1}{n(n-1)}\left[\left(d_{j,0}-1\right)\;c_{j,n-2}+\sum_{k=1}^{\max\left(M,n-2\right)}d_{j,k}c_{j,n-k-2}\right],\;\mathrm{dla}\;n>1$$

An important problem in the presented approach is to find the number of terms of the series (9), which ensures its convergence to the solutions of the Schrödinger equation (see presented in detail further in Section 5). For the coefficients *c_j_*_,0_ and *c_j_*_,1_ any numerical values can be taken. It allows for obtaining different pairs of independent solutions to Equation (2) in the form
(11)$$\Psi_{j}\left(u\right)=C_{j}^{I}F_{j}^{I}\left(u\right)+C_{j}^{\mathrm{II}}F_{j}^{\mathrm{II}}\left(u\right)$$

Assuming
(12)$$c_{j,0}=1,c_{j,1}=i\sqrt{1-d_{j,0}},\;\mathrm{for}\;F_{j}^{I}\left(u\right)$$
(13)$$c_{j,0}=1,c_{j,1}=-i\sqrt{1-d_{j,0}},\;\mathrm{for}\;F_{j}^{\mathrm{II}}\left(u\right)$$
where $i=\sqrt{-1}\;$ and considering that
(14)$$W_{j}\left(u\right)=const=d_{j,0}$$
we get
(15)$$F_{j}^{I}\left(u\right)=\exp\left(i\sqrt{1-d_{j,0}}u\right)\;\mathrm{and}\;F_{j}^{\mathrm{II}}\left(u\right)=\exp\left(-i\sqrt{1-d_{j,0}}u\right)$$

After taking into account the boundary conditions we obtain
(16)$$\left[\begin{array}{c} C_{j+1}^{I}\\ C_{j+1}^{\mathrm{II}} \end{array}\right]=M_{j}\left[\begin{array}{c} C_{j}^{I}\\ C_{j}^{\mathrm{II}} \end{array}\right]$$
where **M***_j_* is the transfer matrix for *j*th layer of the structure in the form
(17)$$M_{j}=\frac{1}{2}\left[\begin{array}{cc} D_{j}^{I}+\beta_{j}G_{j}^{I} & D_{j}^{\mathrm{II}}+\beta_{j}G_{j}^{\mathrm{II}}\\ D_{j}^{I}-\beta_{j}G_{j}^{I} & D_{j}^{\mathrm{II}}-\beta_{j}G_{j}^{\mathrm{II}} \end{array}\right]$$
for matrix elements described by dependencies
(18)$$G_{j}^{K}={\frac{dF_{j}^{K}}{du}|}_{u=\sigma_{j}s_{j}}=\sum_{n=1}^{\infty }nc_{n}\left(\sigma_{j}s_{j}\right){\;}^{n-1},\;\mathrm{for}\;K=I,\;\mathrm{II}$$
and
(19)$$D_{j}^{K}=F_{j}^{K}\left(\sigma_{j}s_{j}\right),\;s_{j}=z_{j+1}-z_{j},\;\beta_{j}=-i\frac{m_{j+1}\sigma_{j}}{m_{j}\sigma_{j+1}}$$
where
(20)$$s_{j}=z_{j+1}-z_{j}\;\mathrm{and}\;\beta_{j}=-i\frac{m_{j+1}\sigma_{j}}{m_{j}\sigma_{j+1}}$$

The transfer matrix **M** for the whole structure can be written as
(21)$$M=\prod_{j=L}^{j=0}M_{j}=\left[\begin{array}{cc} m_{\;1,1} & m_{1,2}\\ m_{2,1} & m_{\;2,2} \end{array}\right]$$

Assuming zero amplitudes of incident waves from the source and drain sides respectively as
(22)$$C_{0}^{I}=0\;\mathrm{and}\;C_{L+1}^{\mathrm{II}}=0$$
we get
(23)$$C_{L+1}^{I}=m_{1,2}\left(E\right)\;C_{0}^{\mathrm{II}}$$
and finally
(24)$$m_{2,2}\left(E\right)=0$$

When solving numerically the Equation (24), the set of self energies of the system is obtained. This provides information on minibands available for electron transport. The task tends to cause problems while simulating specific QCL structures. Therefore, a new computational algorithm (see Figure 4) dedicated to QCL with high energy barriers and narrow quantum wells was developed. The new algorithm is explained in Table 1, where a description for the parameters and procedures is provided. The numerical problems related to the algorithm are described in the following chapters.

Optimising the created algorithm provided an important research problem to solve. Still, with parameters defined as no. 11–14 (see Table 1) and following the details described in Section 4 and Section 5, we managed to succeed.

As shown in Figure 2, depicting the applied algorithm, the Poisson equation is also solved by the QCL simulation process represented with
(25)$$\frac{d}{dz}\varepsilon (z)\frac{dV_{S}(z)}{dz}=-e\rho (z)$$
where *ε*(*z*) is the dielectric permittivity, *e* the elementary charge, and *ρ*(*z*) stands for the charge density function, calculated on the basis of the relationships described in our previous paper [24], and by applying the results obtained from solving the Schrödinger equation. Under the Equation (25) both the potential and its derivative at the boundary of each structure layer must be of continuous character, hence
(26)$$V_{j}\left(z_{j+1}\right)=V_{j+1}\left(z_{j+1}\right)$$
(27)$$\varepsilon_{j}{\frac{dV_{j}}{dz}|}_{z_{j+1}}={\varepsilon_{j+1}\frac{dV_{j+1}}{dz}|}_{z_{j+1}}$$

The charge density function *ρ*(*z*) and the total potential *V_s_*(*z*) in Equation (25) are represented by polynomials in the form [24]
(28)$$\rho_{j}\left(z\right)=\sum_{k=0}^{N}a_{j,k}{\left(z-z_{j}\right)}^{k}$$
and
(29)$$V_{S}{}_{j}\left(z\right)=\sum_{k=0}^{N+2}b_{j,k}^{*}{\left(z-z_{j}\right)}^{k}$$

The polynomial coefficients were calculated as
(30)$$b_{j,k+2}^{*}=\frac{a_{j,k}}{\left(k+1\right)\left(k+2\right)\varepsilon_{j}}$$

The last element of the FMSL (see Figure 2) used in the QCL simulation process is the block for determining Green’s functions. Its role and operating regime were presented in our previous paper [23]. Encouragingly, unlike the procedures for solving the Schrödinger and Poisson equations, it did not require any additional changes. The obtained results for transport parameters were consistent with the results obtained with alternative models, namely WFM and RSM [22,24] applied. Thus, this fragment of the model is not reported in the paper.

## 4. Numerical Analysis of the Function *m*_2,2_(*E*)

Two approaches were applied in the process of solving Equation (24). The first one, used also in [24], consisted in discretisation of specific energy range with an appropriate interval and searching for roots of the *m*_2,2_(*E*) function by the standard bisection method (BM). The second approach (MBM), presented in this paper, engages the complex function *m*_2,2_(*E*) monotony to finding its zeros. Due to large range of values, the tested function was normalised as below
(31)$$N=\sqrt{{\left(\sum_{i=1}^{2}\sum_{j=1}^{2}m_{i,j}(E)\right)}^{2}}$$

It limited the *m*_2,2_(*E*)/N function to the range <−1, 1>, which not only significantly facilitated the analysis of the results, but also accelerated numerical calculations.

Application of standard BM to finding the roots of the *m*_2,2_(*E*) function, with an appropriately adjusted energy step (*dE* = 10^−4^ ÷ 10^−6^ eV), yielded good results for relatively simply superlattice structures, e.g., [24], in terms of obtained results accuracy, and calculations speed. By testing the sign of the function *m*_2,2_(*E*), in the selected energy intervals *dE_k_*, it was possible to iteratively determine its zeros (see Figure 5a).

Applying a similar approach to simulating QCL structures that emit mid and far infrared radiation failed and resulted in errors, which is illustratively explained in Figure 5b Clearly, at the ends of the energy interval *dE_k_* the sign of the function *m*_2,2_(*E*) does not change, so the procedure BM cannot be started. Reducing the energy discretisation to the level as low as *dE_k_* = 10^−7^ eV not only failed, but also impacted negatively the calculations efficiency, as *n*-fold reduction of the *dE_k_* means the procedural time for determining the roots of the *m*_2,2_(*E*) function in the n^2^ dimension has been extended.

The problem was overcome by implementing additional numerical procedures, which were to discretise the tested energy range with variable intervals, based on the monotonicity analysis of the *m*_2,2_(*E*) function. The idea of this approach (MBM) is schematically illustrated in Figure 6. 

Each of the energy intervals *dE_k_* is divided into *N_L_* sections, and then monotonicity is subjected to tests. For example, in the interval <*p*_1_, *q*_1_> the function *m*_2,2_(*E*) keeps decreasing for all linear segments, but in the interval <*p*_2_, *q*_2_> it is not monotonous within the segment *L_k_*_,2_. Then, the discretisation procedure is called recursively and the functions on the non-monotonic section are divided into successive *N_L_* parts. Once the monotonicity of all newly created sections is tested, the program either invokes recursively the discretisation procedure, or proceeds with the examination of the next energy range *dE_k_*. Finally, we obtain a set of energies representing the ends of monotonic sections of the *m*_2,2_(*E*) function, which is later used to find its zeros. Then the sign of the *m*_2,2_(*E*) function at the ends of its monotonic sections is tested, and if the sign difference occurs, the BM procedure starts (see sections *L*_k,1_ i *L*_k,3_).

The effectiveness of the approach presented here was tested by simulating the structure presented elsewhere [28]. The basic parameters of the simulations and selected results (related to method optimisation) of the MBM are presented in Table 2. It should be added that Δ*E* denotes the simulated energy range and the parameter *T_S_* defines the execution time of the loop for solving the Schrodinger and Poisson equations.

The results are summarised in Table 3 where the allowed minibands calculated with BM and MBM are provided. Additionally, Figure 7 shows the *m*_2,2_(*E*) functions calculated with BM (Sym. 1) and the newly developed MBM. The calculations were performed for 4 and 5 periods of the researched QCL structure ((see Sym. 1) and (Sym. 2), respectively). For all the simulations thermodynamic equilibrium conditions and temperature of *T* = 100 K were assumed; simulations were performed on a standard PC with Windows 10 and an Intel core i7 processor.

As shown in Figure 7, the first detected miniband with the BM applied happens also to be the third consecutive miniband resulting from the MBM approach. It means that the results of Sym. 1 (in blue) have not involved two allowed minibands located in the energy ranges of about 0.06 and 0.1 eV that are vital for electron transport (see states marked as *a* and *b* in Figure 1). The picture resulting from with MBM approach looks different, and as shown by the graph of the *m*_2,2_(*E*) function obtained in Sym. 2 (in black) it was able to correctly locate all the allowed minibands within the energy range from 0 to Δ*E_C_*.

The research has shown that in our simulations, the *N_L_* parameter responsible for dividing the energy range *dE_k_* into *L_k_* sections (see Figure 6), where the monotonicity of the *m*_2,2_(*E*) function is tested, to play a vital role. Its value, if underestimated, may result in failing to detect all the function zeros, as it evidently proved true for Sym. 2, where for *N_L_* = 10 in the energy range of 1 mb. (the fragment marked as F in Figure 7a) only 3 out of 5 expected eigenvalues were observed. A very narrow range of this miniband (about 1.3 × 10^−6^ eV) and closely located other roots (even within the order of 1 × 10^−8^ eV) required increased accuracy of the *m*_2,2_(*E*) analysis with regards to its monotonicity. The division of the considered energy range *dE_k_* into 20 sections (*N_L_* = 20) allowed all the expected self energies to be determined accurately.

It is worth noting here that applying MBM with fixed *dE_k_* for the above calculations is faster and more effective than BM with reduced *dE_k_*. For example, the procedure for determining the minibands with *N_L_* = 10 lasted 120 s (duration *T_s_*), while with the parameter *N_L_* increased to 20, *T_s_* = 1470 s was measured. It is a definite progress when compared to the case where the previous version of the model was used and reducing *dE_k_* to 1 × 10^−7^ extended the computation time from 80 s (see Table 2) to about 8000 s, without any capability to effectively detect all the allowed minibands in the structure.

As part of the research, the self energy calculations for the structure [3] were also carried out, and their selected results along with the basic simulation parameters are listed in Table 4. The calculations were run under the same supply and environmental conditions as in Sym. 1–Sym. 3. In the subsequent tasks, the parameter *N_L_* was increased and the obtained results are shown in Table 4. It turned out that the simulations of the structure [3] needed to increase the accuracy of the calculations related to the structure [28]. It was contributed to the higher potential barriers (Δ*E*_c_ = 0.52 eV) and the related occurrence of two very narrow minibands, instead of one. The first one (1 mb.) lies in the range of 1.3 × 10^−6^ eV, while the width of the second one (2 mb.) equals 4.9 × 10^−6^ eV. As shown by the results (see Table 4), setting the *N_L_* parameter to 20, unlike in Sym. 3, did not provide accurate results. Despite the detection of all 12 allowed minibands, the program failed to calculate a complete set of expected self energies, as for 5 periods of the superlattice, 60 were expected. For *N_L_* = 25, the exact values of all self energies were obtained, but as demonstrated, the simulation lasted 5200 s.

It is also worth mentioning here that calculations where thermodynamic equilibrium conditions are assumed require the highest accuracy, as the smallest self energy difference for the case described above was 9.26 × 10^−9^ eV. With voltage applied to the structure, the differences between self energies tend to increase, and consequently, they can be determined much faster.

## 5. Convergence of Solutions to the Schrödinger Equation

The proposed solutions to the Schrödinger equation in the form of polynomials raise the question of the number (*n_k_*) of the sequence terms (9) to be applied. A small number of *n_k_* means faster calculations, though solutions accuracy suffers. A large number of *n_k_* ensures good convergence of the sequence (9) to exact solutions, but it significantly extends computation time. It should be added here that expression (9) is used twice in the process of solving the Schrödinger equation. Primarily it is used to determine the zeros of the *m*_2,2_(*E*) function (Equation (24)), and secondly, their application occurs in wave functions calculating procedure for self energies. In the context of optimising computations, it is worth noting that as solving Equation (24) requires greater number of operations, the process lasts much longer than the one for determining the wave functions.

Initial calculations (Sym. 1 to Sym. 3) assumed *n_k_* = 100 to ensure very accurate results. It is known, however, that simulations of QCL cover many, often time-consuming, processes (see Figure 2), so it is highly advisable to optimise their duration. It prompted the authors to seek solutions for reducing the sequence length (9) as much as applicable in order to accelerate calculations. The analysis of results obtained for the assumed different values of two parameters gave way to proper approach. Namely, the parameter *n_kS_* was responsible for the number of the sequence terms (9) was applied under the process of determining self energies (Equation (24)), while *n_kD_* defined the number of such terms for the wave functions calculations.

Applying voltage to the structure also affects the simulation time. For small voltages, self energies within a given miniband differ slightly, particularly for the allowed minibands that are fairly narrow. For such cases detection of quantum states is difficult, as the simulation process is significantly extended. It is also known that for *U_DS_* = 0, the base of quantum states is determined, as used in NEGF formalism. Thus, this case seems to be a key factor for the model optimisation process. Taking into consideration all the above, the simulation conditions and two main parameters for the evaluation of their results were successfully defined. The accuracy of the calculated self energies was checked on the basis of the error $\delta_{E_{l}^{\nu }}$ value defined as
(32)$$\delta_{E_{l}^{\nu }}=\left|\frac{E_{l}^{\upsilon }-{\tilde{E}}_{l}^{\upsilon }}{E_{l}^{\upsilon }}\right|\cdot 100\%$$
where ${\tilde{E}}_{l}^{\nu }$ is the *l*th value of the calculated self energy within the miniband *ν* according to sequence (9) of length *n_kS_*, while $E_{l}^{\nu }$ is the value of the same self energy calculated for *n_kS_* = 100. The maximum value $Max\left|\delta_{E_{l}^{\nu }}\right|$ calculated for *l* self energies was taken as representative for the miniband *ν*.

The accuracy of the calculated wave functions was determined in a similar way, i.e., by comparing them to the result obtained at *n_kD_* = 100. In this case, the error parameter was defined according to the relationship
(33)$$\delta_{\Psi_{l}^{\nu }}=\sqrt{\frac{{\int \left(\Psi_{l}^{\upsilon }-{\tilde{\Psi }}_{l}^{\upsilon }\right)}^{2}dz}{\int {\left(\Psi_{l}^{\upsilon }\right)}^{2}dz}}\cdot 100\%$$
where ${\tilde{\Psi }}_{l}^{\upsilon }$ is the *l*th value of the calculated wave function within the miniband *ν* according to sequence (9) of length *n_kD_*, and $\Psi_{l}^{\upsilon }$ is the value of the same wave function calculated for *n_kD_* = 100. The maximum value $Max\left|\delta_{\Psi_{l}^{\nu }}\right|$ calculated for *l* self energies was taken as representative for the miniband *ν*.

Five modules (*L_p_* = 5) of QCL structures provided the basis for analysing the obtained results. The assumed number of modules (periods) corresponds to the conclusion reported in our previous paper [23], where it was found to be the minimum size of the FMSL that maximises location of quantum states, and has been found to be an effective base for the NEGF formalism. Another important aspect of the current research was also the tested energy range. We focused solely on the first and eighth minibands (represented by states *b* and *c* in Figure 1) due to their fundamental importance for the electron transport in the considered structures. The complexity degree of the respective wave functions that differ significantly has been also considered important for selecting minibands to be analysed. Wave functions representing 8 miniband are much more complicated than those related to 1 miniband. Therefore, most likely the *n_kD_* number for miniband *c* needs to be increased to approximate correctly the wave functions; numerical studies were to answer whether it was necessary.

The semi-analytical approach to QCL modelling allows for calculating the values of the wave functions with high accuracy, without significantly affecting the calculations performance. It is so basically due to the *n_Z_* parameter, i.e., the number of calculation points for one layer of the structure. 

Selected results of the calculations for the first miniband are illustrated in Figure 8 and listed in Table 5, where the basic parameters of the simulations can also be found. The table contains data from four simulations (Sym. 8–Sym. 11) carried out for different values of parameters related to the length of the sequence (9) (*n_kS_*, *n_kD_*).

All the plots collectively shown in Figure 8 present the calculated real parts of the wave functions (Re Ψ*^b^*) under thermodynamic equilibrium conditions. The calculations were performed within MBM approach for *N_L_* = 20, in the energy range Δ*E* = 0.06 ÷ 0.07 eV with the basic discretisation step *dE* = 1 × 10^−6^ [eV]. 

The results pictured in Figure 8a show that too small number (*n_kS_* = *n_kD_* = 20) of terms of the sequence (9), despite sufficiently precise discretisation of the energy band (*N_L_* = 20), failed to determine all eigenstates in the miniband *a*. Hence, we observed here only one state with the energy *E*_1_ = 0.06019953 eV instead of the expected five quantum states.

Moreover, the wave function representing the considered state does not converge to the assumed zero in the source and drain regions (see fragments F1 and F2). The maximum error ($Max\left|\varepsilon_{\Psi_{l}^{b}}\right|$) of convergence of the wave function calculated according to the Formula (33) is 72.76%, whereas the maximum self energy error ($Max\left|\delta_{E_{l}^{b}}\right|$) for this case equalled 9.81%. Such a large error $Max\left|\varepsilon_{\Psi_{l}^{b}}\right|$ in the source and drain areas results not only from the high $Max\left|\delta_{E_{l}^{b}}\right|$ value, but also from relatively wide electrode zone (10 nm was assumed here) in relation to the thickness of other module layers (see Table 2). As the values of the polynomial coefficients are determined at the layers’ boundaries, moving away from them when calculating the wave functions, leads to increased inaccuracies.

By increasing the *n_kS_* and *n_kD_* parameters to 25, we eliminated the failure to detect all the eigenstates, and reduced the error practically to zero ($Max\left|\delta_{E_{l}^{b}}\right|$ = 1.5 × 10^−10^%), though good convergence of the wave functions at the ends of the simulated structure remained to be perfected. This is shown by the results of Sym. 10 in Figure 8b, namely, their fragment marked as F, where $Max\left|\varepsilon_{\Psi_{l}^{b}}\right|$ = 2.81%.

Setting the *n_kD_* parameter to the value of 100, as shown in Figure 8c, with the same value of *n_kS_* produced correct results for the miniband *a* and good convergence of the calculated wave functions within the whole structure ($Max\left|\varepsilon_{\Psi_{l}^{b}}\right|$ = 0.21%). Correct normalisation of the wave function in the area of the whole structure and the accurate calculation of the related electron charge were possible to be executed in the next stage. It is also worth noting that the parameter selection, proposed in Sym. 11, only slightly increased the computation time from 70 to 78 s, but resulted in nearly six-fold acceleration with respect to the same computations in Sym. 8.

The results for the series (9) convergence with the solutions of the Schrödinger equation in the miniband (c) are shown in Figure 9 and listed with the simulation parameters in Table 6. They include simulations of 5 periods of the QCL [28] under thermodynamic equilibrium with MBM for *N_L_* = 20, for the energy range Δ*E* = 0.23 ÷ 0.24 eV with the basic discretisation step *dE* = 1 × 10^−6^ [eV].

It turned out that setting the parameters *n_kS_* and *n_kD_* to 20 in Sym. 13, unlike Sym. 9, allowed for detecting all the expected self energies in the miniband under consideration. The maximum error in the self energy value ($Max\left|\delta_{E_{l}^{c}}\right|$), of only 3.5 × 10^−7^ eV occurred here. Regretfully, no good convergence of the function to the assumed zero in the area of the drain, marked as F in the picture, was found and the maximum error ($\varepsilon_{\Psi_{l\max}^{c}}$) was of 1.55%. This error was minimised to the value of 2.1 × 10^−4^% by setting the parameters *n_kS_* and *n_kD_* to 25, while the error value $Max\left|\delta_{E_{l}^{c}}\right|$ was reduced to 2.4 × 10^−10^ eV (see Sym. 14). It all contributed to accurate computation of wave functions and eigenvalues in a very short time of 8 s. It should be added here that the results of Sim. 14 are so close to the results of Sim. 12 that they were not presented in the Figure 9.

It is then reasonable to conclude that despite more complex waveforms of the wave functions in the 8 miniband in relation to 1 miniband, it proved unnecessary to increase the value of the *n_kD_* parameter in order determine quickly and accurately all the eigenstates for the tested structure.

The width of the allowed miniband seems to remain the decisive factor with regard to optimising the concerned method. For the concerned case it was relatively large and amounted to 1.89 meV. The subsequent simulations results that were carried out for the structure reported in [3] confirmed such findings. In Table 7 all the basic parameters and selected results of the above calculations carried out within the range of 2 miniband under thermodynamic equilibrium conditions are listed.

The presented results show that for Sym. 16 and Sym. 17 the errors $Max\left|\delta_{E_{l}^{b}}\right|$ exceeded the value of the allowed miniband width (4.9 × 10^−6^ eV). Hence, the software found only one self energy and the wave function for this energy could not be determined as the sequence (9) was divergent (*n*.c.). Setting the *n_kS_* and *n_kD_* parameters to 30 (see Sym. 18) allowed all the expected self energies to be detected and the accuracy of the calculations was increased to the level $Max\left|\delta_{E_{l}^{\nu }}\right|$ = 1.0 × 10^−7^. A very small error $Max\left|\varepsilon_{\Psi_{l}^{b}}\right|$ = 9.3 × 10^−3^ required no additional increase for the *n_kD_* parameter. In relation to Sym. 15 nearly four times shorter simulation time was achieved.

## 6. Conclusions

Contemporary QCL structures emitting mid- and far-infrared radiation typically contain very narrow quantum well systems that tend to generate discrete self energies which are difficult to detect. Hence, numerical models dedicated to such devices need to smartly combine computations that are equally high accurate and efficient. The proposed approach allows a variable discretisation step to be applied with regards to the structure geometrical dimensions and the considered energy range alike. Due to semi-analytical nature of the method, with an appropriate set of parameters for the developed algorithm, simulation process is found optimal, which classifies the FMSL model among highly desirable and effective tools, supportive in designing QCL.

## Figures and Tables

**Figure 1 materials-15-05715-f001:**
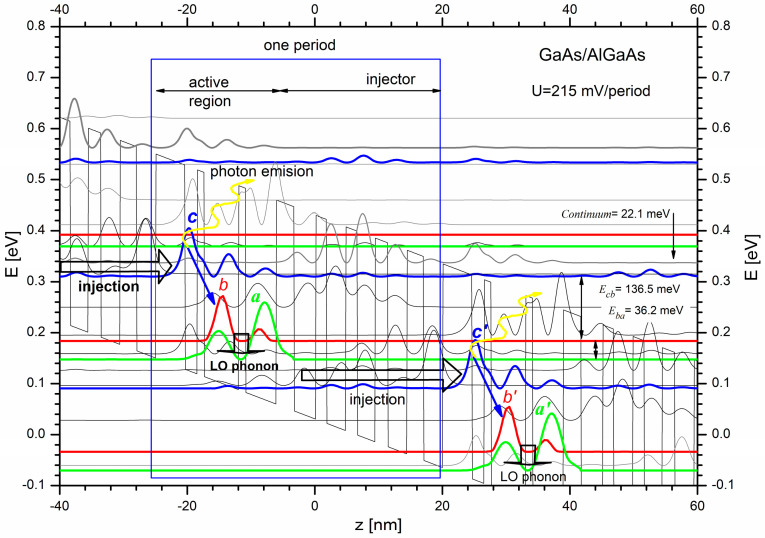
Illustration of the basic mechanisms for the QCL. The electrons tunnel from the relaxation (injection) region of the previous structure period to the high state (**c**) of the next superlattice period, followed by a transition to the medium state (**b**) with a photon emission. In the next step, the electron goes to the low state (**a**) by emitting a phonon and due to the electric field, it is further transported through the injection area to the high state of the next period (**c’**), where the sequence of transitions between states is repeated. Namely, we observe the photon transition between states (**c’**) and (**b’**) and then the emission of the phonon after the electron transition from state (**b’**) to (**a’**).

**Figure 2 materials-15-05715-f002:**
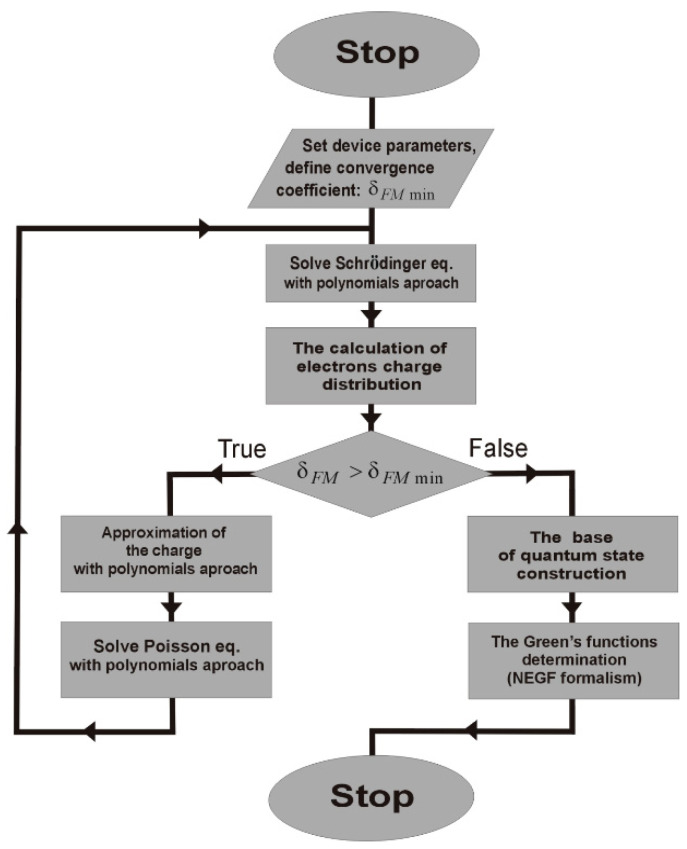
The main algorithm for self-consistent solving the Schrödinger and Poisson equations in the process of QCL simulations.

**Figure 3 materials-15-05715-f003:**
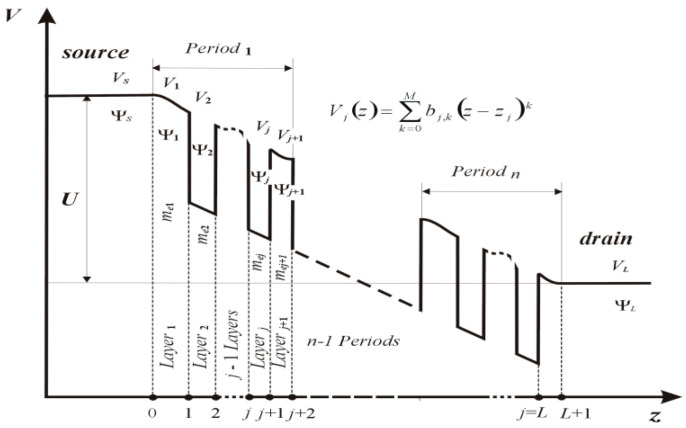
Concepts of polynomial approximation of the potential in the QCL structure. In each superlattice layer, a constant value of the effective mass of the electron and a variable potential described by the polynomial function *V_j_*(*z*) were assumed.

**Figure 4 materials-15-05715-f004:**
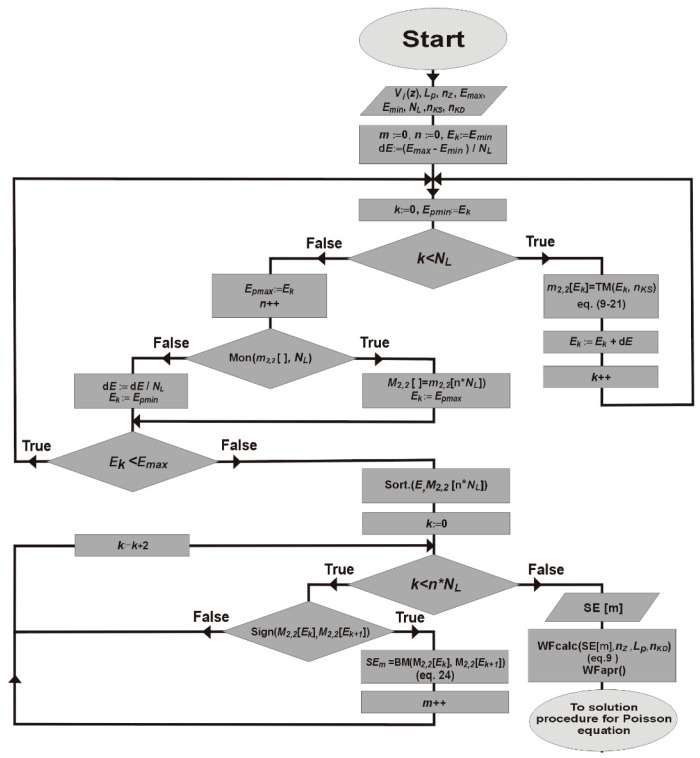
The algorithm of the Schrödinger equation solving process under the new approach to determining self energy.

**Figure 5 materials-15-05715-f005:**
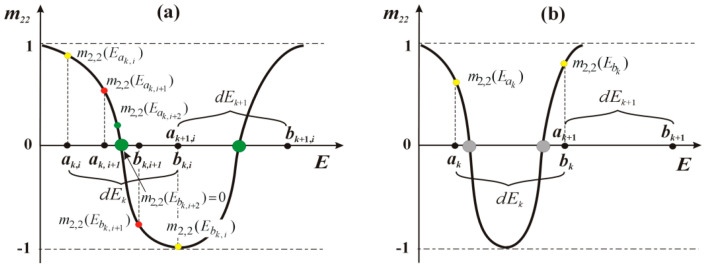
Schematic illustration of the standard BM engaged in the process of finding the zeros of the *m*_2,2_(*E*) function: (**a**) an appropriately selected energy discretisation step *dE*_k_ ensures the sign of the function at the ends of the *dE*_k_ = <*a_k_*, *b_k_*> segment to be changed and the specific segment to include the root, which can be found in the next iterative step (*i* + 2); (**b**) for *dE*_k_ values that are too large the sign of the *m*_2,2_(*E*) function does not change and the zeros are not found.

**Figure 6 materials-15-05715-f006:**
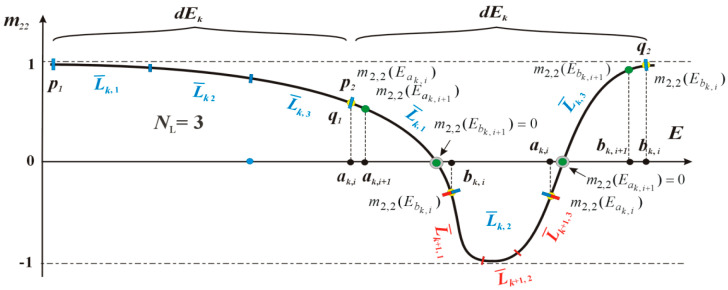
A schematic illustration of the new approach (MBM) to finding zeros of the *m*_2,2_(*E*) function. After dividing the interval <*p*_1_, *q*_1_>, into *N_L_* parts, all the sections are monotonic; hence, the procedure can take the next energy interval <*p*_2_, *q*_2_>. The non-monotonicity of the section *L_k_*_,2_ triggers recursively its division into *N_L_* parts. After dividing the *m*_2,2_(*E*) function into monotonic sections, the sign at the ends of each section is checked and the procedure BM is started if signs are found to differ.

**Figure 7 materials-15-05715-f007:**
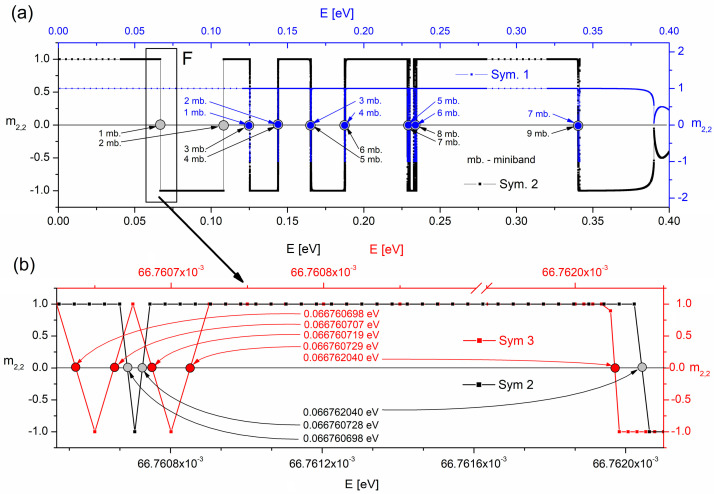
Graphs of the *m*_2,2_(*E*) function obtained in the simulation of the structure reported in [28]: (**a**) for 4 periods (Sym. 1) and 5 periods (Sym. 2–3) of the superlattice, with BM (Sym. 1) and MBM (Sym. 2–3) applied, with the interval *dE* = 1^−6^ eV; (**b**) calculation results of the *m*_2,2_(*E*) function for the part marked in (**a**) as F with parameters: *N_L_* = 10 (Sym. 2) oraz *N_L_* = 20 (Sym. 3).

**Figure 8 materials-15-05715-f008:**
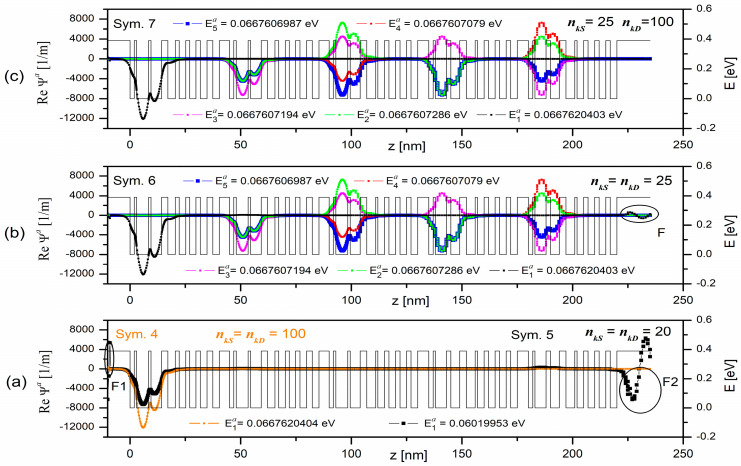
Illustration of the series (9) convergence with the solutions of the Schrödinger equation (Re Ψ*^b^*) depending on the degree of the approximating polynomial represented by the parameters *n_kS_* and *n_kD_*. The calculations were performed for the first minibandof the structure consisting of 5 QCL periods reported in [28]. Basic simulation parameters are included in Table 5. Graph (**a**) shows the wave functions (Re Ψ*^b^*) calculated for the parameters *n_kS_* = *n_kD_* = 20, graph (**b**) is the same wave functions calculated for the parameters *n_kS_* = *n_kD_* = 25, and graph (**c**) illustrates the above wave functions calculated for the parameters *n_kS_* = *25 and n_kD_* = 100.

**Figure 9 materials-15-05715-f009:**
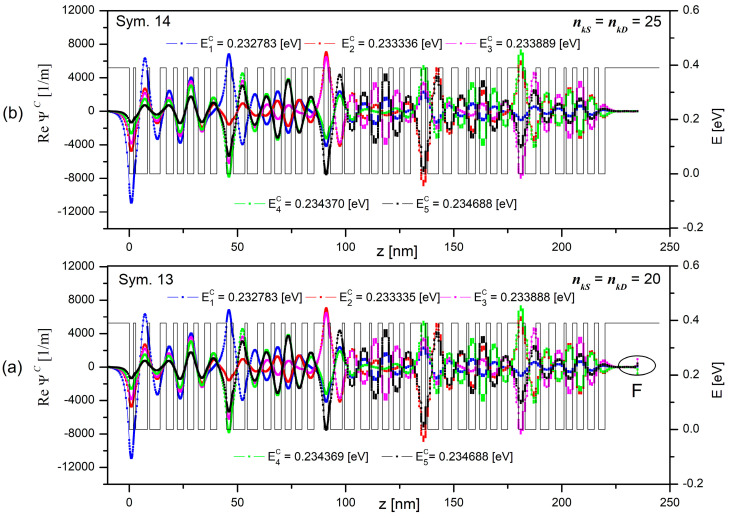
Convergence of the series (9) with the solutions to the Schrödinger equation (Re Ψ*^c^*) depending on the degree of the approximating polynomial *n_k_*. The calculations were performed for the eighth miniband of the structure which consists of 5 QCL periods as described in [28]. Basic simulation parameters are listed in Table 6. Graph (**a**) shows the wave functions (Re Ψ*^c^*) calculated for the parameters *n_kS_* = *n_kD_* = 20, graph (**b**) is the same wave functions calculated for the parameters *n_kS_* = *n_kD_* = 25.

**Table 1 materials-15-05715-t001:** Basic parameters and numerical procedures of the Schrödinger equation solving algorithm (see Figure 4).

Parameter	Denoted Physical Quantity/Notion	Unit
1. *Lp*	The number of the structure periods	-
2*. Vj*(*z*)	The total potential in the QCL structure	[eV]
3*. k*	Index of the energies currently being considered	
4. *n*	Index of the monotonicity vector for the *m*_22_(*E*) function	
5. *m*	Index of the self energies	
6. *E_min_*	The minimum of the assumed energy range	[eV]
7. *E_max_*	The maximum of the assumed energy range	[eV]
8. *dE*	Interval of the assumed energy range	[eV]
9. *SE* [m]	Self energies vector	[eV]
10. *m*_22_ [*N_L_*]	*m*_2,2_(*E*) function vector	
11. *N_L_*	The number of parts the *dE_k_* interval is dived into under MBM procedure	-
12. *n_KS_*	The number of terms in series (Equation (9)) under the self energy calculation procedure	-
13. *n_KD_*	The number of terms in series (Equation (9)) under the wave function calculation procedure	-
14. *n_z_*	The number of the grid points in a single layer of the structure	-
**Procedure**	**Denoted Procedure**	
TM (*E_k_, n_KS_*)	Transfer matrix procedure	
Mon (*m*_22_ [ ], *N_L_*)	Monotonicity test procedure	
Sort (*E*, M_22_ [*n***N_L_*])	Sorting procedure for the *m_2,2_(E*) arguments (ascending order)	
Sign (M_22_[*E_k,_*], M_22_[*E_k+_*_1_])	Procedure for checking the sign of a *m*_2,2_*(E)*	
BM (M_22_[*E_k,_*], M_22_[*E_k+_*_1_])	Bisection method procedure	
WFcalc (SE[m], *L_p_*, *n_z_*, *n_KD_*)	Wave functions calculation procedure	
WFapr ()	Wave functions approximation procedure	

**Table 2 materials-15-05715-t002:** Basic parameters and selected simulation results for structure presented in [28].

*Simulation Parameters*								*Selected Results*		
Sym. No.	Sym. Meth.	*Lp*	Structure Layers [nm]	Δ*E*_c_	Δ*E*	d*E*	*N*_*L*_ (MBM)	Expect. Self En. No.	Calc. Self En. No.	*T*_*S*_ [s]
				[meV]						
Sym. 1	BM	4	**4.6**, 1.9, **1.1**, 5.4, **1.1**, 4.8, **2.8**, 3.4, **1.7**, 3.0, **1.8**, 2.8, **2.0**, 3.0, **2.6**, 3.0	390	1 ÷ 400	1 × 10^−3^	-	36	28	80
Sym. 2	MBM	5					10	45	43	120
Sym. 3							20		45	1470

**Table 3 materials-15-05715-t003:** Allowed minibands calculated for structure reported in [28] by using the BM method (Sym. 1) and MBM method (Sym. 2 and Sym. 3).

Sym. No.	1 mb.	2 mb.	3 mb.	4 mb.	5 mb.	6 mb.	7 mb.	8 mb.	9 mb.
				[meV]					
Sym. 1	125.393 ÷ 125.431	143.827 ÷ 143.961	165.172 ÷ 165.479	187.473 ÷ 187.856	228.578 ÷ 230.212	232.791 ÷ 234.631	340.359 ÷ 341.420		
Sym. 2 Sym. 3	66.760 ÷ 66.762	108.237 ÷ 108.246	125.390 ÷ 125.431	143.827 ÷ 143.961	165.172 ÷ 165.479	187.473 ÷ 187.856	228.578 ÷ 230212	232.791 ÷ 234.631	340.359 ÷ 341.420

**Table 4 materials-15-05715-t004:** Basic parameters and selected simulations results for structure presented in [3].

*Simulation Parameters*								*Selected Results*		
Sym. No.	Sym. Meth.	*Lp*	Structure Layers [nm]	Δ*E*_c_	Δ*E*	d*E*	*N* _ *L* _	Expect. Self En.	Calc. Self En.	*T*_*S*_ [s]
				[meV]				No.		
Sym. 4	MBM	**5**	**4.0**, 1.9, **0.7**, 5.8, **0.9**, 5.7, **0.9**, 5.0, **2.2**, 3.4, **1.4**, 3.3, **1.3**, 3.2, **1.5**, 3.1, **1.9**, 3.0, **2.3**, 2.9, **2.5**, 2.9,	520	1 ÷ 520	1 × 10^−3^	10	60	51	235
Sym. 5							15		52	373
Sym. 6							20		52	2400
Sym. 7							25		60	5200

**Table 5 materials-15-05715-t005:** Basic parameters and selected results of simulations for structure presented in [28].

*Simulation Parameters*								*Selected Results*				
Sym. No.	Δ*E*	d*E*	*N* _ *L* _	*Lp*	*n* _ *Z* _	*n* _ *kS* _	*n* _ *kD* _	Exp. Self En.	Calc. Self En.	$\mathrm{Max}\;\;\left|\delta_{E_{l}^{c}}\right|$	$\mathrm{Max}\;\;\left|\delta_{\Psi_{l}^{c}}\right|$	*T*_*S*_ [s]
	[meV]							No.		[%]		
Sym. 8	60 ÷ 70	1 × 10^−3^	20	5	20	100	100	5	5	-		420
Sym. 9						20	20		1	9.81	72.76	50
Sym. 10						25	25		5	1.5 × 10^−7^	16.14	70
Sym. 11						25	100		5		0.21	78

**Table 6 materials-15-05715-t006:** Basic parameters and selected results of simulations for structure presented in [28].

*Simulation Parameters*								*Selected Results*				
Sym. No.	Δ*E* [eV]	d*E* [eV]	*N* _ *L* _	*Lp*	*n* _ *Z* _	*n* _ *kS* _	*n* _ *kD* _	Expect. Self En. No.	Calc. Self En. No.	$\mathrm{Max}\;\;\left|\delta_{E_{l}^{c}}\right|\;\;[\%]$	$\mathrm{Max}\;\left|\delta_{\Psi_{l}^{c}}\right|\;\;[\%]$	*T*_*S*_ [s]
Sym. 12	0.23 ÷ 0.24	1 × 10^−6^	20	5	20	100	100	5	5	-		41
Sym. 13						20	20		5	3.5 × 10^−7^	1.55	6
Sym. 14						25	25		5	2.4 × 10^−10^	2.1 × 10^−4^	8

**Table 7 materials-15-05715-t007:** Basic parameters and selected results of simulations for structure presented in [3].

*Simulation Parameters*									*Selected Results*				
Sym. No.	Δ*E* [eV]	d*E* [eV]	*N* _ *L* _	*Lp*		*n* _ *Z* _	*n* _ *kS* _	*n* _ *kD* _	Exp. self en. no.	Calc. self en. no.	$\mathrm{Max}\;\;\left|\delta_{E_{l}^{b}}\right|\;\;[\%]$	$\mathrm{Max}\;\;\left|\delta_{\Psi_{l}^{b}}\right|\;\;[\%]$	*T*_*S*_ [s]
Sym. 15	0.1 ÷ 0.101	1 × 10^−6^	25	5	20		100	100	5	5	-	-	1787
Sym. 16							20	20		1	2.7 × 10^−3^	n. c.	5
Sym. 17							25	25		1	2.3 × 10^−5^	n. c.	7
Sym. 18							30	30		5	1.0 × 10^−7^	9.3 × 10^−3^	450

## Data Availability

Not applicable.

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
