# Peer review of "Optimisation of QCL Structures Modelling by Polynomial Approximation"

_materials, 2022, doi:10.3390/ma15165715_

Round 1
Reviewer 1 Report
Reviewer report:
The manuscript: Optimisation of QCL Structures Modeling by Polynomial Approximation is interesting work in the field of quantum cascade laser modeling. The main question addressed in the manuscript is the applicability of the polynomial approximation method for solving Schrodinger equation. A new algorithm for determining self energies has been proposed and a new method has been optimized. The Paper is well-written text is clear and easy to read. Conclusions are consistent with obtained results.
I recommend publishing the manuscript after addressing the following comments:
1 Authors claim that it is vital to increase model accuracy to deal with very narrow minibands (beginning of page 2). Minibands of the order of 10^-6 eV (page 10, after fig. 7; page 11, before table 4) are given as examples later in the text. It is necessary to give more argumentation of why such narrow minibands need to be modeled. An example of a superlattice of certain materials, certain layer thickness, and certain accuracy of layer thickness fabrication should be given. The possibility of fabrication of such a superlattice (with a mini bandwidth of 10^-6 eV) should be defensible. It can turn out that the required accuracy of fabrication is more than a monolayer.
2 Manuscript contains many sections and is difficult to navigate. Including a description of manuscript content at the end of the introduction section will help.
3 Figure 1 is overloaded. Better to show one or two periods.
4 Figure 7, the X-axis scale and values are confusing. Better to change to meV. Also, give values in meV in tables 2 and 3.
5 Table 2 gives structure layer width with 0.1 nm accuracy (see 1.1 nm, 1.9 nm…). The given accuracy of 0.1 nm can not be higher than monolayer thickness, which is typically 0.2 nm. An example should be realistic. Please see my comment 1 again.
6 What is DeltaE (column 6 of table 2)? Is it miniband width? It is not defined in the text or in the parameters table (table 1).
Author Response
- Ans. 1.
An except has been added on page 3:
For the correct operation of the device, it is necessary to ensure the population inversion for high quantum states, at a level that allows cascading photon emissions in subsequent periods of the structure. Unfavourable physical phenomena causing the escape of electrons into the energy continuum makes it much more complex. Therefore, the first cascade lasers worked properly only at cryogenic temperatures [29]. The solution to the problem was achieved by appropriately configuring two [30] or three [31] quantum wells, or even complete superlattices [32]-[33] within the active area of the laser structure. By introducing a very thin quantum well, followed by two wider quantum wells in the active area of the laser, we have managed to increase significantly the efficiency of injecting carriers to the upper level (c) and to reduce the effect of harmful transfer of carriers directly to the lower quantum level (a). All these solutions generate numerous quantum states, often very similar in energies and grouped in very narrow mini-bands. They can be determined by self-consistent solving the Schrödinger and Poisson equations, though it remains quite a difficult task. The problem can be solved in several ways, one of which is described in the following chapter.
- Ans. 2.
The description has been completed in the form:
As the paper deals with several quite separate issues, we divided it into following sections:
- Chapter 2 where operation of QCL with a typical structure emitting radiation in the mid-infrared range is briefly described,
- Chapter 3 where we present the numerical QCL model into which a new algorithm for precise self-energy search has been incorporated, and its basic procedures are described,
- Chapter 4 where validity of implementing such changes into the current QCL model is discussed and supported with relevant calculation results.
And last but not least
- Chapter 5 where the convergence of solutions to the Schrödinger equation is analysed, which is of key importance for model optimisation, as confirmed with relevant examples.
The performed studies finally has led to creating an efficient and accurate QCL model dedicated to systems with very narrow quantum wells and high potential barriers for electronic transport.
- Ans. 3.
The figure has been corrected.
- Ans. 4.
All changes have been made.
- Ans. 5.
A fair remark, but on the other hand, it should be remembered that there is interface roughness in the real structure. Therefore, we consider the thickness dimension rather as an average value. This makes it possible to determine the initial position of the quantum state in the well after solving the Schrodinger equation. The effect of electron scattering on the interface roughness is taken into account in the calculation of the Green’s functions. When determining the thickness of the layers in specific structures, the publications that write them are also important, on which we base the simulation parameters.
- Ans. 6.
An except has been added on page 10
It should be added here that ΔE denotes the simulated energy range and the parameter TS defines the execution time of the loop for solving the Schrodinger and Poisson equations.

Reviewer 2 Report
The manuscript overall is of good quality. A very comprehensive study. I recommend its acceptance with very minor corrections.
A brief overview of QCL can be added with more references.
Follow the same patterns in references.
The conclusion needs to be rechecked
Author Response
1. A brief overview of QCL can be added with more references.
Ans. The note has been taken into account by referring to additional literature.
2.Follow the same patterns in references.
Ans. The note was taken into account by making corrections in the reference list.
The conclusion needs to be rechecked
Ans. The conclusions have been redrafted.

Reviewer 3 Report
This manuscript reports on improvements in accuracy and efficiency of a numerical methods to model finite quantum cascade laser structures. The authors introduce and optimize a novel algorithm to solve both the Schrödinger and Poisson equations using polynomial approximations for the finite superlattice structure. The critical parameters for accurately extracting the self-energies and all the minibands relevant for transport in the structures are identified and the efficiency of the method for a realistic structure is tested and compared to the previous algorithm developed by the authors.
The accurate and efficient modeling of QCL structures is essential for the understanding and design of these widely used lasers. The topic of research is revelant for a journal such as Materials. I found the paper very much on the technical side and strongly focused on the particular method developed by the authors, but, as the authors clearly explain and illustrate their numerical optimization procedure, I believe that it may have added values for researchers working on these specific modeling issues. I would thus give a positive recommendation for publication in Materials.
Author Response
Ans. The positive opinion of the reviewer was read with joy and further steps were taken towards the publication of the work.
Reviewer 4 Report
Dear Editor,
About the manuscript :
Title: Optimisation of QCL Structures Modeling by Polynomial Approximation.
Authors: Stanisław Pawłowski and Mariusz Mączka.
In this work, the authors investigated the modeling of quantum cascade laser (QCL) structures, despite a regular progress in the field, still remains a complex task in both analytical and numerical aspects. In particular contemporary QCL structures, with quantum wells quite typically forming complex systems, a special approach to determining self energies and coefficients of approximating polynomials is required. Polynomial approximation method has been successfully applied to solving the Schrödinger equation in QCLh. Important results have been obtained and investigated. I think that this manuscript is important and merit publication in MDPI Materials ,
1) Introduction part should contained research papers/latest/related or research results should be cited to improve scientific quality of manuscript such as the following works:
https://doi.org/10.1016/j.spmi.2017.09.020
https://doi.org/10.1016/j.spmi.2017.12.060
2) Question: Instead of general conclusion, the authors need to optimize their findings for the technology specific where their findings could be employed along with merits for the same over the existing device architecture.
Author Response
Ans.1) Changes have been made to the introduction and the reference list to reflect the above note
Ans. 2) Changes have been made to the content of the conclusion to reflect the above note
